# Metformin Ameliorates 2.856 GHz Microwave- Radiation-Induced Reproductive Impairments in Male Rats via Inhibition of Oxidative Stress and Apoptosis

**DOI:** 10.3390/ijms241512250

**Published:** 2023-07-31

**Authors:** Junqi Men, Li Zhang, Ruiyun Peng, Yanyang Li, Meng Li, Hui Wang, Li Zhao, Jing Zhang, Haoyu Wang, Xinping Xu, Ji Dong, Juan Wang, Binwei Yao, Jiabin Guo

**Affiliations:** 1PLA Center for Disease Control and Prevention, Beijing 100071, China; menjunqi1@163.com (J.M.); zhangli_526@163.com (L.Z.); 2Institute of Radiation Medicine, Academy of Military Medical Sciences, Beijing 100850, China; ruiyunpeng18@126.com (R.P.); lyy547152380@163.com (Y.L.); lm17732588219@163.com (M.L.); wanghui597bj@163.com (H.W.); lillyliz@163.com (L.Z.); zhang115614@163.com (J.Z.); smart106@126.com (H.W.); xxpbjhd@163.com (X.X.); djtjwj@163.com (J.D.); annie341393@126.com (J.W.); 3School of Public Health, China Medical University, Shenyang 110122, China

**Keywords:** microwave radiation, metformin, male reproductive damage, oxidative stress, apoptosis

## Abstract

The reproductive system has been increasingly implicated as a sensitive target of microwave radiation. Oxidative stress plays a critical role in microwave radiation -induced reproductive damage, though precise mechanisms are obscure. Metformin, a widely used antidiabetic drug, has emerged as an efficient antioxidant against a variety of oxidative injuries. In the present study, we hypothesized that metformin can function as an antioxidant and protect the reproductive system from microwave radiation. To test this hypothesis, rats were exposed to 2.856 GHz microwave radiation for 6 weeks to simulate real-life exposure to high-frequency microwave radiation. Our results showed that exposure to 2.856 GHz microwave radiation elicited serum hormone disorder, decreased sperm motility, and depleted sperm energy, and it induced abnormalities of testicular structure as well as mitochondrial impairment. Metformin was found to effectively protect the reproductive system against structural and functional impairments caused by microwave radiation. In particular, metformin can ameliorate microwave-radiation-induced oxidative injury and mitigate apoptosis in the testis, as determined by glutathione/-oxidized glutathione (GSH/GSSG), lipid peroxidation, and protein expression of heme oxygenase-1 (HO-1). These findings demonstrated that exposure to 2.856 GHz microwave radiation induces obvious structural and functional impairments of the male reproductive system, and suggested that metformin can function as a promising antioxidant to inhibit microwave-radiation-induced harmful effects by inhibiting oxidative stress and apoptosis.

## 1. Introduction

With the rapid development of electrical equipment and wireless transmission technology, microwave devices have been increasingly used in daily life and industries such as telecommunications, medical devices, and military weapons, causing increasing concern for human health and environmental safety. Since the first report by Wertheimer showing the impact of electromagnetic radiation on fertility [1], significant efforts have been made to investigate the effects of microwave radiation on reproductive functions following occupational and environmental exposure [2]. Compared to other tissues, the testis with spermatogenic cells has been found to be more sensitive to microwave radiation, as the testis has lesser tissue protection and is more easily penetrated by microwave radiation [3,4]. According to epidemiological surveys and experimental studies, microwave radiation can damage testicular function, semen quality, and testicular structure [5,6,7].

The microwave band ranges in frequency from 300 MHz to 300 GHz [8], but the majority of studies of microwave-radiation-induced reproductive impairment have focused on the frequencies of 900 MHz, 1800 MHz and 2.45 GHz [7,9,10,11]. Emitters with a frequency of 2.856 GHz have been widely used in radar and other communication devices [12]. Our previous studies and other reports showed spatial memory and cognitive impairment and dose-dependent cardiomyocyte injury in rats, calcium efflux in hippocampal neurons and cardiomyocytes, and potential cytotoxicity in bone marrow mesenchymal stem cells following 2.856 GHz microwave exposure [13,14,15,16,17,18,19]. However, few studies have investigate the potential deleterious effects and the underpinning mechanisms of 2.856 GHz microwave exposure on reproduction, especially after long-term exposure.

Emerging evidence suggests that oxidative stress is critically involved in the pathogenesis and development of microwave radiation-induced damage to the reproductive system. It has been revealed that microwave radiation increases reactive oxygen species (ROS) in spermatozoa, which can deteriorate antioxidant enzymes and induce mitochondrial dysfunction, eventually damaging male fertility [20,21,22,23]. Under oxidative conditions, spermiogenesis becomes defective and defective gametes form, leading to unusual apoptosis and reproductive impairment [24].

Metformin, a biguanide compound widely used for type 2 diabetes mellitus, has been increasingly shown to be a promising radioprotective agent. Karam et al. reported that metformin protected against cardiac damage caused by ionizing-radiation-induced oxidative stress, inflammatory mediators, and endothelial dysfunction [25]. However, the precise effects of metformin on a reproductive system exposed to microwave radiation are unclear.

Metformin can stimulate cellular antioxidant enzymes, to indirectly scavenge free radicals and decrease lipid peroxidation, as well as to modulate cellular metabolism and mitochondrial function [26,27,28,29]. Recently, emerging evidence suggests that metformin protection is closely associated with its antioxidant capacity in the reproductive system. Metformin was found to increase sperm concentration and sperm viability, enhance antioxidant enzyme activity, and improve testicular tissue sperm quality [30,31]. These studies suggest that metformin may be a useful countermeasure against microwave radiation, but to date the effects of metformin on microwave radiation have not been investigated.

We established an animal model of microwave-radiation-induced damage that is closer to real-life exposure and demonstrated that metformin attenuated 2.856 GHz microwave-radiation-induced alteration in rats’ sperm motility and structure, serum hormones, and mitochondrial function. Our results suggest that metformin can function as a promising antioxidant in inhibiting microwave-radiation-induced oxidative stress and apoptosis and in protecting male fertility.

## 2. Results

### 2.1. Metformin Relieved the Decrease in Sperm Motility Caused by Microwave Radiation

Sperm motility is one of the most widely used indicators to reflect reproductive function. Progressive sperm motility was defined as grade A + B; as shown in Figure 1, the proportion of grade A + B sperm in the radiation group was significantly lower than in the control group at 6 h, 7 d, and 14 d after microwave radiation (*p* < 0.01). However, in the metformin groups, the proportions of grade A + B sperm were significantly higher compared to the radiation group in a dose-dependent manner (*p* < 0.05 or *p* < 0.01) (Figure 1A). There was no statistical difference in the proportion of nonprogressive sperm (grade C) between the groups at all four time points after microwave radiation (Figure 1B). The proportion of immotile sperm (grade D) showed the opposite results to grade A + B sperm. Compared with the control group, the proportion of grade D was increased at 6 h, 7 d, and 14 d after microwave radiation (*p* < 0.01). After metformin administration, the proportion of grade D was decreased dose-dependently compared with the radiation group (*p* < 0.05 or *p* < 0.01) (Figure 1C). Collectively, the results showed that microwave radiation impaired sperm motility, and metformin increased sperm motility in radiated rats in a dose-dependent manner.

### 2.2. Metformin Attenuated the Abnormal Testicular Structure Induced by Microwave Radiation

The basic pathological changes in the rat testis after exposure to microwave radiation were characterized by loosening of the germinal epithelium, disorganization of the spermatogenic cells, vacuolation, degeneration, necrosis, and interstitial edema [9,32]. In our experiment, the most significant changes were observed 6 h after exposure to 2.856 GHz microwave radiation, and abnormalities of the testicular structure were reduced after metformin administration. As shown in Figure 2, the histological structures of the testicular tissue of the control rats were normal and clear. Compared to the control group, seminiferous tubules with loose arrangement and spermatogenic cells with vacuolation, disorganization, necrosis, degeneration, and desquamation were observed after microwave radiation. The pathological changes of testicular tissue in the R + L group were similar to those in the radiation group, but to a lesser degree. In the R + M group, a loosening of the spermatogenic epithelium and interstitial edema of the testicular tissue were observed. In the R + H group, only occasional loosening of the spermatogenic epithelial arrangement of the testicular tissue was observed.

Ultrastructural changes in the testis were evaluated by electronic microscopy. In spermatogonia, spermatocytes and spermatids, cytoplasmic swelling and cavitation, chromatin condensation, mitochondrial swelling and hollowing were observed 6 h after microwave radiation, and these ultrastructural alterations were significantly alleviated by metformin. As shown in Figure 3, the ultrastructure of testicular tissue of the control group rats was normal. After radiation, large edema of spermatogonia, apoptotic necrosis of spermatocytes, edema of spermatids and swollen and cavitated mitochondria were observed. In the R + L group, condensation and border shift of sperm chromatin, spermatocytes and spermatids edema, and mitochondrial swelling and cavitation were observed. The damage was significantly reduced in the R + L and R + H groups, but was least severe in the high-dose group.

### 2.3. Metformin Ameliorated the Microwave-Radiation-Induced Alteration of Hormones

Testosterone and Inhibin B have been frequently used as hormone biomarkers indicating reproductive functions. Our results showed that the testosterone level in the radiation group was significantly higher than in the control group at 6 h, 7 d, 14 d, and 28 d after radiation (*p* < 0.01), and metformin reversed the changes in radiated rats (*p* < 0.05 or *p* < 0.01) (Figure 4A). A significant decrease in Inhibin B was observed in the microwave radiation group at 6 h, 7 d, 14 d, and 28 d (*p* < 0.01); furthermore, in the microwave radiation + metformin groups, the level was significantly higher compared to the radiation group at 6 h, 7 d, 14 d, and 28 d after radiation (*p* < 0.01), with the most significant effect in the R + H group (*p* < 0.05 or *p* < 0.01) (Figure 4B). The results showed that the concentrations of testosterone and Inhibin B were significantly altered by microwave radiation, and metformin could dose-dependently inhibit the alterations.

### 2.4. Metformin Ameliorated the Microwave-Radiation-Induced Reduction in Energy Metabolism Enzymes

Normal reproductive function is inseparable from important enzymes in energy metabolism. LDH, SDH, and α-GLU are sperm-specific energy metabolism enzymes involved in spermatogenesis and energy metabolism. Among these enzymes, LDH is a key enzyme in glycolysis [33] and SDH is an indispensable enzyme in oxidative phosphorylation [34]; they are widely distributed and found in germ cells and are a crucial source for energy metabolism [35]. α-GLU is secreted by the epididymis, which catalyzes glucose catabolism and gives energy to sperm maturation and sperm motility [36,37]. Our results showed that exposure to microwave radiation reduced the content of LDH and SDH enzymes at 6 h, 7 d, 14 d, and 28 d and α-GLU enzyme at 6 h and 7 d after microwave radiation (*p* < 0.05 or *p* < 0.01, respectively), compared with the control group. Compared with the microwave radiation group, the LDH and SDH content increased in the microwave radiation + metformin groups at 6 h, 7 d, 14 d, and 28 d (*p* < 0.05 or *p* < 0.01), and the α-GLU content increased in the microwave radiation + metformin groups at 6 h, 7 d, and 28 d after radiation, especially in the radiation plus the high dose metformin group (*p* < 0.05 or *p* < 0.01) (Figure 5A–C).

### 2.5. Metformin Alleviated Oxidative Stress Damage Caused by Microwave Radiation

Oxidative stress is one of the major mechanisms of microwave radiation-induced damage to the body. To determine whether the mechanism by which metformin alleviates microwave radiation-induced damage is associated with its antioxidant property, the levels of GSH and MDA and the protein expressions of HO-1 in the testes of rats were determined. Compared to the control group, exposure to microwave radiation caused significant decreases in the GSH/GSSG ratio at 6 h, 7 d, and 14 d and increases in the MDA level at the four points after radiation (*p* < 0.05 or *p* < 0.01, respectively). Compared with the microwave radiation group, the changes induced by microwave radiation were reversed by metformin in a dose-dependent manner (*p* < 0.05 or *p* < 0.01, respectively) (Figure 6A,B). Heme oxygenase-1 (HO-1) is a major factor of antioxidation. We investigated the antioxidative effect of metformin on HO-1 expression. Our results showed that the protein expression of HO-1 was down-regulated in the radiation group at the four points after microwave radiation but increased in microwave radiation + metformin groups (*p* < 0.05 or *p* < 0.01, respectively) (Figure 6C,D). In summary, exposure to microwave radiation could cause oxidative damage to reproductive organs, and metformin could act as an antioxidant to reduce this damage.

### 2.6. Metformin Mitigated Microwave-Radiation-Induced Mitochondrial Impairments

Mitochondria are a major target of microwave radiation-induced oxidative stress. Mitochondria-related enzymes can reflect mitochondrial function and the degree of oxidative stress damage [38]. Our results showed that the contents of mitochondrial complex IV and ATP synthase at the four points were reduced by microwave radiation (*p* < 0.01). Compared to the microwave radiation group, these deleterious effects were mitigated by metformin in a dose-dependent manner (*p* < 0.05 or *p* < 0.01, respectively) (Figure 7A,B).

### 2.7. Metformin Mitigated Microwave-Radiation-Induced Apoptosis

The Bax/Bcl-2 ratio (the apoptotic index) reflected the vulnerability of the tissue to apoptosis. In our experiment, Bax/Bcl-2 expression was up-regulated in the radiation group at 6 h, 7 d, and 14 d, and the changes were reversed after the administration of metformin (*p* < 0.05 or *p* < 0.01, respectively) (Figure 8A,B). The expression of cleaved caspase-3 was up-regulated in the radiation group at 6 h, 7 d, 14 d, and 28 d. Compared to the microwave radiation group, the expression of cleaved caspase-3 was down-regulated in microwave radiation + metformin groups (*p* < 0.05 or *p* < 0.01, respectively) (Figure 8A,C). The increase in the Bax/Bcl-2 and cleaved caspase-3 expression suggested that apoptosis could be caused by microwave radiation and that metformin inhibited this phenomenon.

## 3. Materials and Methods

### 3.1. Animals

A number of 100 8-week-old male Wistar rats with a weight of 200 ± 20 g were supplied by Beijing Vital River Laboratory Animal Technology Co., Ltd. (Beijing, China), and kept in ventilated specific pathogen-free (SPF) facilities. The rats were kept in a separate room with a controlled temperature of 23 ± 1 °C, a relative humidity of 55 ± 10%, and a 12 h light/dark light cycle. Ad libitum food and distilled water were provided. The laboratory animals were handled according to the US National Institutes of Health Guide for the Care and Use of Laboratory Animals. All protocols were approved by the Institutional Animal Care and Use Committee (IACUC-AMMS-2020–780).

### 3.2. Microwave Radiation and Animal Treatments

The rats were randomly divided into five groups of 20 rats each, including the control group (C), in which rats were administered normal saline via gastric gavage without microwave radiation exposure; the microwave radiation group (R), in which rats were exposed to microwave radiation and administered normal saline via gastric gavage; the microwave radiation plus 10 mg/kg metformin group (R + L), in which rats were exposed to microwave radiation and administered 10 mg/kg metformin via gastric gavage; the microwave radiation plus 30 mg/kg metformin group (R + M), in which rats were exposed to microwave radiation and administered 30 mg/kg metformin via gastric gavage; and the microwave radiation plus 90 mg/kg metformin group (R + H), in which rats were exposed to microwave radiation and administered 90 mg/kg metformin via gastric gavage.

The rats were exposed to microwave radiation at a radiation frequency of 2.856 GHz and a mean power density of 30 mW/cm^2^ for 6 w, 5 d/w, and 15 min/d. The microwave generator, a klystron amplifier model JD 2000 (Vacuum electronics research institute, Beijing, China) was able to generate pulsed microwaves at 2.856 GHz. Metformin was administered by gavage at 10, 30, and 90 mg/kg b.w for 21 consecutive days from 14 days before the last exposure to 7 days after the completion of radiation. After 6 h, 7, 14, and 28 days of radiation, the animals were anesthetized with 1% sodium pentobarbital (0.5 mL/100 g) by intraperitoneal injection and dissected. Blood, testes, and epididymis were collected. The testis was frozen in liquid nitrogen or fixed in a 4% formaldehyde solution. The epididymis was frozen or collected for the detection of sperm motility. The principle of microwave radiation is shown in Figure 9A. Figure 9B shows the radiation apparatus and experimental design.

According to the latest ICNIRP 2020 guidelines for limiting electromagnetic field (EMF) exposure, EMFs below 6 GHz can penetrate tissues and are usually described by the SAR. The SAR values of each rat testicular tissue during microwave radiation were calculated using the finite difference in time domain (FDTD) method, which was reported in our previous study [39]. Hence, the SAR value of 30 mW/cm^2^ microwave radiation to testicular tissue in this experiment was 34.2 W/kg, and the whole-body average SAR value for rats was 10.17 W/kg.

### 3.3. Test of Sperm Motility

To evaluate spermatozoa motility, a computer-assisted semen analysis on a—TOX IVOS II^TM^ CASA System (Hamilton Thorne, PA, USA) was used. Sperm was collected and diluted in a physiological solution. Subsequently, the sperm was pipetted onto a counting slide (2 chamber slides, depth 100 μm, Barcelona, Spain) and was immediately analyzed. A minimum of five counting chamber fields per measurement were evaluated using the IVOS CASA system. At least 1000 sperm per sample were evaluated. Sperm motility was classified as rapid progressive sperm (grade A), sluggish progressive sperm (grade B), nonprogressive sperm (grade C), and immotile sperm (grade D).

### 3.4. Hematoxylin and Eosin Staining

A 10% neutral buffered formalin solution was used to fix the testis for at least one week prior to HE staining. Approximately 3 mm thick testicular tissues were dehydrated, rendered transparent, and boiled in wax before being embedded in paraffin. The sections (3 μm) were dried for 48 h at 60 ° C and dewaxed in water before being dipped in hematoxylin (Sinopharm Chemical Reagent Beijing Co., Ltd., Beijing, China) for 15 min and washed in tap water for 5 min. Subsequently, the sections were dipped in eosin (Sinopharm Chemical Reagent Beijing Co., Ltd.) for 15 s, then washed again for 5 min. After dehydration of the alcohol gradient, the clearance with xylene and placement of a cover slip were performed. Observations were conducted using the LEICA DM6000 light microscope (Leica, Wetzlar, Germany).

### 3.5. Ultrastructure of Testicular Tissue

The testicular tissues were freshly collected and cut into pieces (1 mm^3^ in size), which were fixed in 2.5%glutaraldehyde. A sequential process was carried out, including incubation with 1% osmium tetroxide, and graded ethyl alcohol, and embedding of the samples in EPON618 (TAAB Laboratories Equipment, Berks, UK). Lead citrate and uranyl acetate were used for staining (Advanced Technology & Industrial Co., Ltd., Hong Kong, China). The slides were visualized by transmission electron microscopy (TEM) (H-7800, Tokyo, Japan).

### 3.6. Detection of Serum Hormones

After anesthetization, blood was collected through the main abdominal vein to prepare the serum. Serum testosterone and Inhibin B were measured using a sensitive immunoradiometric assay, with commercial kits (RK-179, RK-204, Beijing Furui Runze Biotechnology Co., Ltd., Beijing, China) and a radioimmunoassay system (XH6080, Xi’an Nuclear Instrument Co., Ltd., Xian, China). Both testosterone and Inhibin B had intra- and inter-assay coefficients of variation (CV) of 10% and 15%, respectively.

### 3.7. Detection of Sperm-Specific Energy Metabolism Enzymes

The testicular and epididymal tissues were lysed in a lysis buffer and then centrifuged at 5000× *g* for 5 min. The supernatants were collected and used to measure the levels of the energy metabolism enzymes lactate dehydrogenase (LDH), succinodehydrogenase (SDH), and α-glucosidase (α-GLU). The levels of LDH and SDH in testicular tissues and α-GLU in epididymal tissues were measured using assay kits according to the manufacturer’s instructions (Jianglai Biotechnology Co., Ltd., Shanghai, China). Absorbance was measured using an enzyme-linked immune absorbent assay (ELISA) (Rayto, RT-6100, Shenzhen, China).

### 3.8. Evaluation of Oxidative Stress and Mitochondrial Enzymes

The levels of reduced glutathione (GSH), oxidized glutathione (GSSG), and malondialdehyde (MDA) were detected to evaluate oxidative stress in testicular tissues. The level of GSH was determined using a commercial kit (G263, DOJINDO, Kyushu Island, Japan) according to the manufacturer’s protocol. GSSG was selectively quantified by detecting the color reaction after the addition of DTNB (5,5′-dithiobis (2-nitrobenzoic acid)) and the circulating system under the action of glutathione reductase. The concentration of MDA was determined using a commercial kit (MAK085, SIGMA, St. Louis, MO, USA) based on the reactivity of thiobarbituric acid (TBA).

The levels of mitochondrial complex IV and ATP synthase were measured using assay kits (Jianglai Biotechnology Co., Ltd., Shanghai, China). Enzyme levels were positively correlated with the shade of the substrate TMB color. Absorbance was measured by enzyme-linked immune absorbent assay (ELISA) (Rayto, RT-6100, Shenzhen, China).

### 3.9. Expression of Proteins in Testicular Tissue

The testicular tissues were lysed in a tissue lysis buffer containing protease inhibitors. The supernatants were collected by centrifuging at 4 °C, 12,000 r/min for 10 min. Total proteins were sampled and subjected to SDS-PAGE. After electrophoresis, membranes were blocked and incubated with primary antibodies against Bcl-2 (ab196495, Abcam, Cambridge, UK), Bax, caspase3 (14796, 9662, CST, Boston, Massachusetts, USA), Heme oxygenase-1 (HO-1), and anti-β-actin (10701, proteintech, San Diego, California, USA; GB11001, Servicebio, Wnhan, China) overnight. Subsequently, the primary antibodies were washed three times with TBST and then incubated with the secondary antibody (7074, CST, Boston, Massachusetts, USA). Bands were semi-quantitatively analyzed using Image J 1.48v (National Institutes of Health, Bethesda, MD, USA).

### 3.10. Statistical Analysis

Data analysis was performed using SPSS 25.0 and statistical charts were made using Prism. Quantitative data were expressed as mean ± S.D. A one-way ANOVA and LSD post hoc test were performed to analyze the data. Statistical significance was designated at *p* < 0.05.

## 4. Discussion

The harmful effects of microwave radiation on the reproductive system have received increasing attention since the Scientific Committee on Emerging and Newly Identified Health Risks (SCENIHR) proposed the potential health effects of electromagnetic field exposure in 2015 [40]. It is well known that the biological effects of microwave radiation are associated with the intensity and duration of exposure and can accumulate in a dose and time-dependent manner [41]. In particular, long-term exposure to high-frequency and high-power microwave radiation may cause dysfunction in various organs, including reproductive organs [13,42]. Although great efforts have been made to discover and develop pharmacological interventions against microwave radiation, effective agents are still absent. In the present study, rats were exposed to 2.856 GHz microwave radiation, a frequency commonly used in many fields such as communications and military, with a mean power density of 30 mW/cm^2^ for 6 weeks. To observe the effects of long-term radiation on the reproductive system and whether metformin has a protective effect, we examined four time points after radiation. Our results showed that microwave radiation induced serum hormone disorders, decreased sperm motility, depleted sperm energy, caused abnormalities in testicular structure, and caused mitochondrial impairment. Metformin, a potential antioxidant, was used as a protective agent against microwave radiation and was found to effectively decrease oxidative stress and apoptosis induced by 2.856 GHz, and eventually improve reproductive impairments.

Sperm motility is indispensable for normal procreation; optimal sperm motility is a benchmark for successful fertilization, and poor sperm motility is a principal contributor to male infertility [43,44,45]. Many studies have reported the negative influence of microwave radiation on sperm motility and morphology. For example, declines in sperm motility and normal sperm morphology were observed in men’s semen due to the increased use of cell phones with frequency bands between 400 MHz and 2 GHz [46]. Others found that sperm motility was reduced in rats and human semen after exposure to microwave radiation frequency of 900 and 850 MHz [47,48]. Furthermore, exposure to 2.45 GHz microwave radiation has been found to induce structure disorder and spermatogenic cell degeneration in the testis [33,49]. Our findings showed that sperm motility was reduced by 2.856 GHz microwave radiation, as evidenced by the decreased proportion of progressive sperm motility and the altered structure of testicular tissues at the cellular and subcellular levels. Metformin, an oral drug that acts systemically, has been shown to improve testis function and semen quality [30]. Furthermore, metformin in a dose-response manner improved drug-induced rat testicular toxicity by increasing sperm count and motility, as well as improved testicular morphology [50]. In our study, metformin administration was found to dose-dependently ameliorate 2.856 GHz microwave-radiation-induced sperm motility reduction and improved morphological alterations. Our findings are in agreement with previous reports showing that metformin may act as a protective agent to protect testicular function and sperm motility.

Testosterone and Inhibin B are two main sex hormones in men. Testosterone is a testicular hormone secreted by Leydig cells [20]. Studies report that testosterone plays an important role in the spermatogenesis process, spermatozoa development, and maintenance of the structure and physiology of the seminiferous tubules [2]. Therefore, testosterone fluctuations can negatively affect male fertility. Microwave exposure disrupts the Leydig cell and affects serum testosterone levels. When exposed to microwave radiation, the seminiferous tubules and their Leydig cells are disrupted, resulting in changes in testosterone levels. Inhibin B is secreted by Sertoli cells, which nurture spermatids [51] and initiate spermatogenesis [52]. As the key endocrine marker of spermatogenesis, the decrease in Inhibin B indicates an impairment of spermatogenesis and a decrease in sperm quality [53]. Continuous exposure to 900 MHz microwave radiation was found to decrease Inhibin B content and reduce sperm motility in rats [9,54]. Consistent with these findings, our results showed an increase in testosterone and a decrease in the Inhibin B content from exposure to 2.856 GHz microwave radiation at all four time points. Metformin has been shown to protect male gonadal function from compensatory function [55]. A recent study showed that metformin can alleviate abnormalities in testosterone secretion [30]. In addition, metformin was also shown to modulate serum Inhibin B [56,57]. Our study showed that metformin attenuated the disturbance of testosterone and Inhibin B hormone by 2.856 GHz microwave radiation to the normal levels, whether they had been increases or decreases.

Testes produce and mature sperm in the epididymis, and a large amount of energy is consumed in the process. Furthermore, sperm motility also requires a sufficient energy supply [58]. In the present study, the levels of the sperm-specific energy metabolism enzymes SDH, LDH, and α-GLU were detected to observe changes in sperm energy metabolism after exposure to 2.856 GHz microwave radiation; these alterations were ameliorated by metformin. Previous studies suggested that melatonin prevents microwave-radiation-induced oxidative damage by increasing the levels of testicular LDH-X [59]. Mice exposed to 3 Gy X-rays exhibited reduced activities of testis enzymes such as SDH and LDH [60]. Consistent with these studies, the activities of SDH, LDH, and α-GLU were weakened after exposure to 2.856 GHz microwave radiation. Whereas few studies have investigated the effect of metformin on sperm-specific energy metabolism enzymes, our results demonstrated that metformin alleviated the inactivity of sperm energy enzymes induced by 2.856 GHz microwave radiation, supporting the reproductive protective effect of metformin in microwave-radiation-induced energy insufficiency in testicular spermatogenic cells.

Although the mechanisms by which metformin exerts its protective effects are poorly understood, increasing evidence suggests that metformin may function as an antioxidant in response to oxidative damage. A growing body of evidence suggests that the protective effects of metformin against radiation are primarily attributed to its antioxidant properties [61]. Metformin has been implicated as an endogenous antioxidant to regulate the redox status of the reproductive system, scavenge excess free radicals, improves antioxidant enzyme activity, and reduces MDA levels [62,63,64]. The antioxidant property of metformin on the reproductive system has also been tested by increasing study. It has been found that metformin stimulated alanine production, induced antioxidant activity, and maintained the equilibrium of NADH/NAD^+^ in Sertoli cells [56]. Others found that metformin improved oxidative stress in germ cells after testicular torsion and improved sperm quality [31]. Additionally, emerging evidence suggests the critical role of oxidative stress in the pathogenesis and development of microwave-radiation-induced damage to the reproductive system [2,4]. Due to the essential role of oxidative stress in microwave-radiation-induced damage and the antioxidative properties of metformin, we examined the effect of metformin of anti-radiation. However, few studies focus on the effect of metformin on microwave radiation protection. Our results showed that exposure to 2.856 GHz microwave radiation decreased GSH/GSSG and HO-1 expression, and increased MDA content in rats, showing significant damage from oxidative stress. Our results also showed that mitochondrial complex IV and ATP synthase content were reduced by 2.856 GHz microwave radiation, which further supports the oxidative damage of microwave radiation. Interestingly, all of these changes were modifiable by metformin, suggesting that metformin exerts an effectively beneficial effect on the deleterious effects of microwave radiation through the possible mechanism of inhibition of oxidative damage.

Metformin not only acts as a free radical scavenger to reduce free radical production, but also reduces radiation-induced apoptosis [65]. In the reproductive system, metformin has been shown to reduce germ-cell-specific oxidative-stress-induced apoptosis that improves sperm quality after testicular damage [31]. Moreover, many studies have shown that microwave radiation induces free radical formation and excessive ROS accumulation in testicular tissue, causing activation of apoptosis [24,66,67]. In the present study, we used Bax/Bcl-2 and cleaved caspase-3 to detect apoptosis in testicular tissues. The results revealed that the Bax/Bcl-2 ratio and the expression of the cleaved caspase-3 were increased by 2.856 GHz microwave radiation exposure, and the increases were inhibited by metformin, indicating an important role for metformin against microwave-radiation-induced apoptosis.

In conclusion, our results demonstrated that exposure to 2.856 GHz microwave radiation for 6 weeks caused obvious structural and functional impairments in the reproductive system and that rats still had not fully recovered from reproductive damage 28 days after radiation exposure, but that metformin protected against reproductive impairments, at least in part by inhibiting oxidative stress and apoptosis. The mechanisms and sensitive targets need to be further elucidated by sequencing, knockdown, or overexpression experiments.

## Figures and Tables

**Figure 1 ijms-24-12250-f001:**
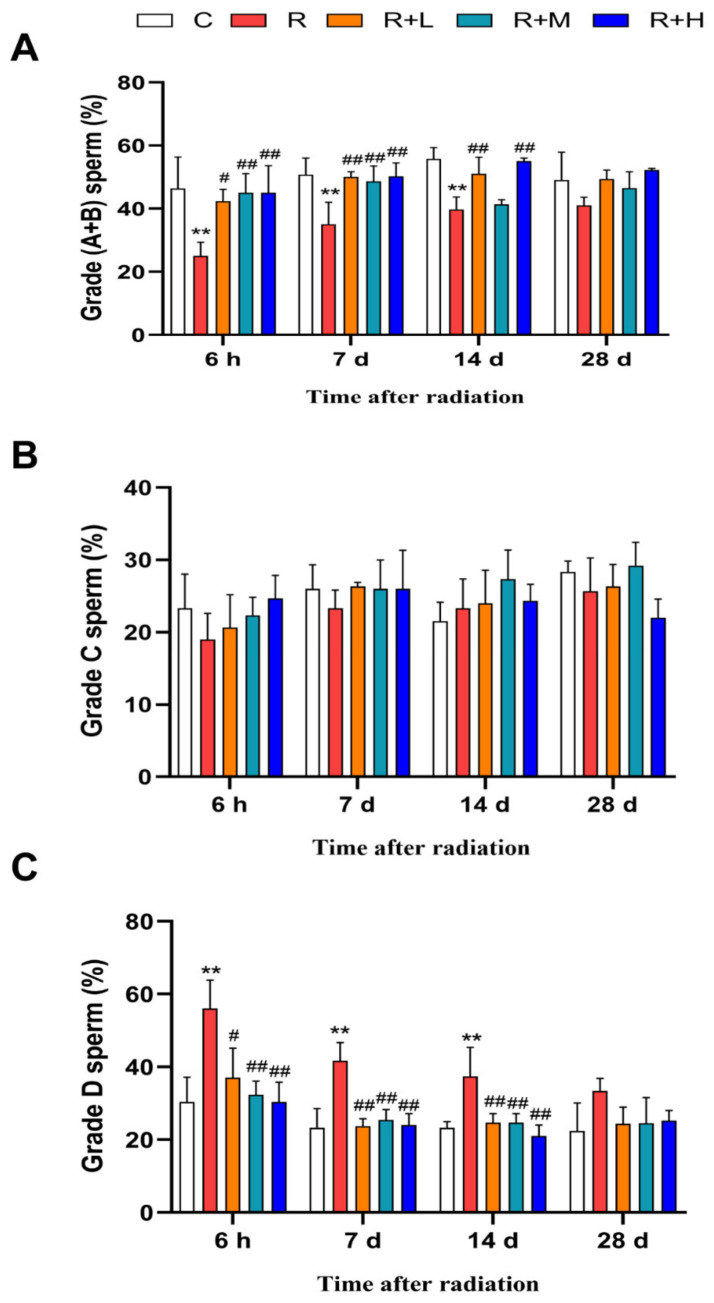
**Figure 1.** Metformin relieved the decrease in sperm motility caused by microwave radiation (n = 5 for each group). (**A**) Changes in the proportion of grade A + B sperm. (**B**) Changes in the proportion of grade C sperm. (**C**) Changes in the proportion of grade D sperm. ** shows *p* < 0.01 for the R group compared to the C group; and # shows *p* < 0.05 for the R + L, R + M, and R + H groups compared to the R group, and ## shows *p* < 0.01.

**Figure 2 ijms-24-12250-f002:**
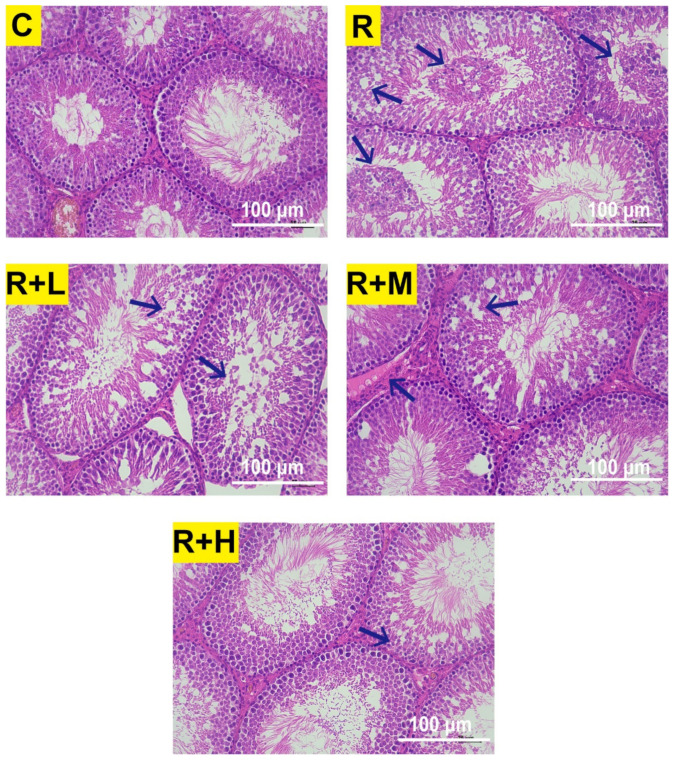
Metformin alleviated the abnormalities of the testicular structure caused by microwave radiation (HE staining, scale bar = 100 μm, n = 5 for each group). In the C group, normal testicular tissues were observed. The R group showed loosened structure of seminiferous tubules, disorderly arrangement, vacuolation, necrosis, degeneration and desquamation of spermatogenic cells at 6 h after microwave radiation. The abnormalities were reduced in the R + L, R + M, and R + H groups, and the R + H group had the lowest number of abnormalities. The arrows show the sites of the injuries.

**Figure 3 ijms-24-12250-f003:**
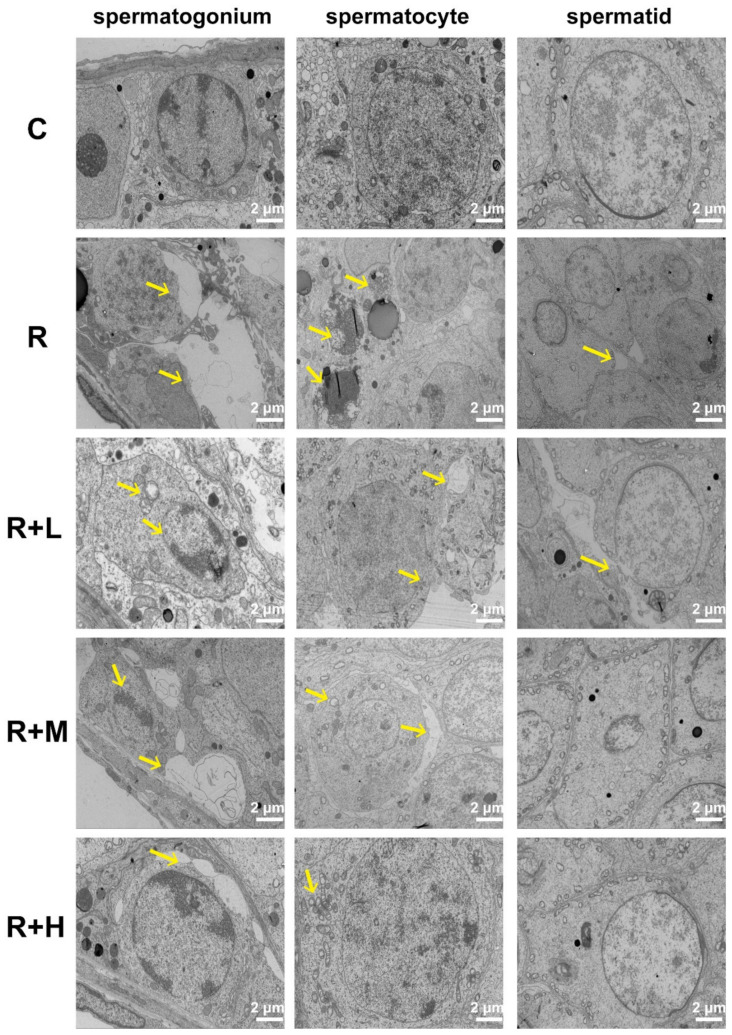
Metformin relieved ultrastructural abnormalities in testicular tissue caused by microwave radiation (TEM, scale bar = 2 μm, n = 3 for each group). Ultrastructural changes of each group at 6 h after microwave radiation are shown. Normal spermatogonia, spermatocytes, and spermatids in testicular tissue were observed in the C group. In the R group, swelling and cavitation, chromatin condensation, mitochondrial swelling, and hollowing were observed. The abnormalities were reduced in the R + L, R + M, and R + H groups; the R + H group had the fewest abnormalities. Arrows indicate the sites of the injuries.

**Figure 4 ijms-24-12250-f004:**
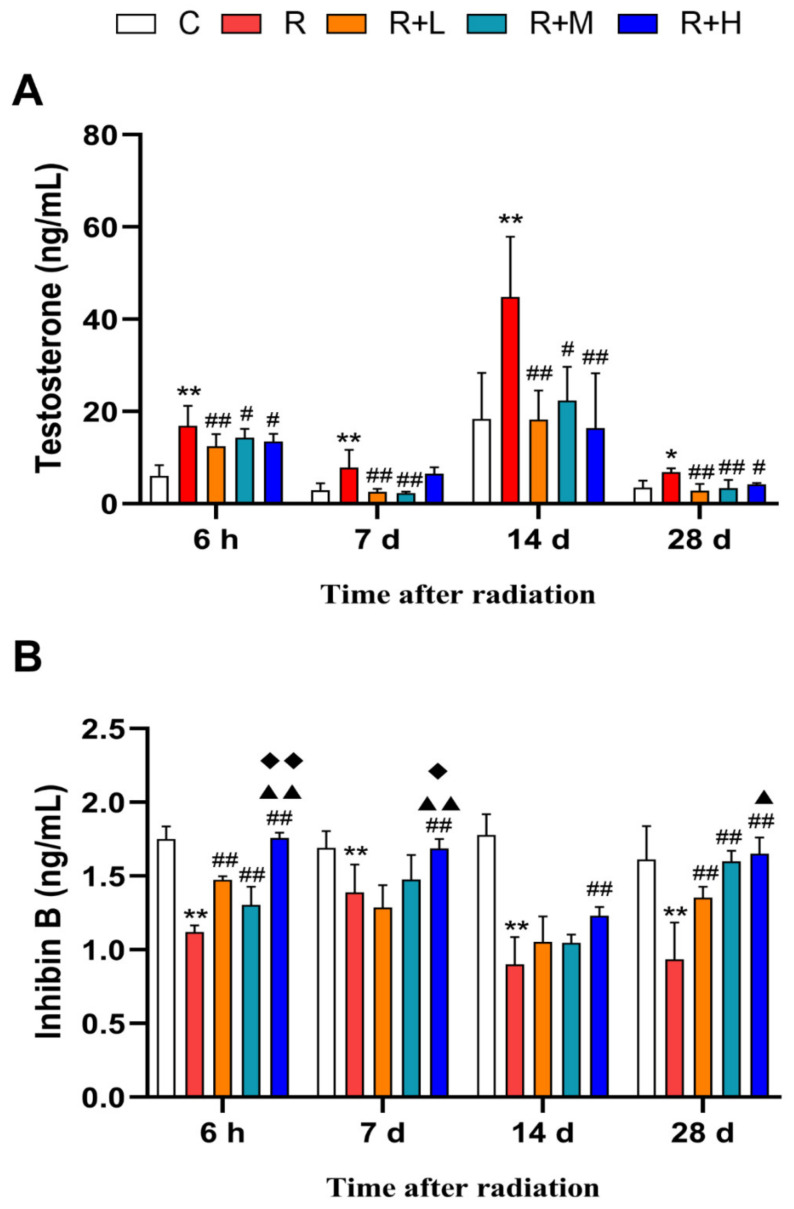
Metformin ameliorated the microwave-radiation-induced alteration of hormones (n = 5 for each group). (**A**) Changes in serum testosterone concentrations. (**B**) Changes in serum Inhibin B concentrations. * shows *p* < 0.05 for the R group compared to the C group, and ** shows *p* < 0.01; # shows *p* < 0.05 for the R + L, R + M, and R + H groups compared to the R group, and ## shows *p* < 0.01; ▲ shows *p* < 0.05 for the R + M and R + H groups compared with the R + L group, and ▲▲ shows *p* < 0.01; and ◆ shows *p* < 0.05 for the R + H group compared to the R + M group, and ◆◆ shows *p* < 0.01.

**Figure 5 ijms-24-12250-f005:**
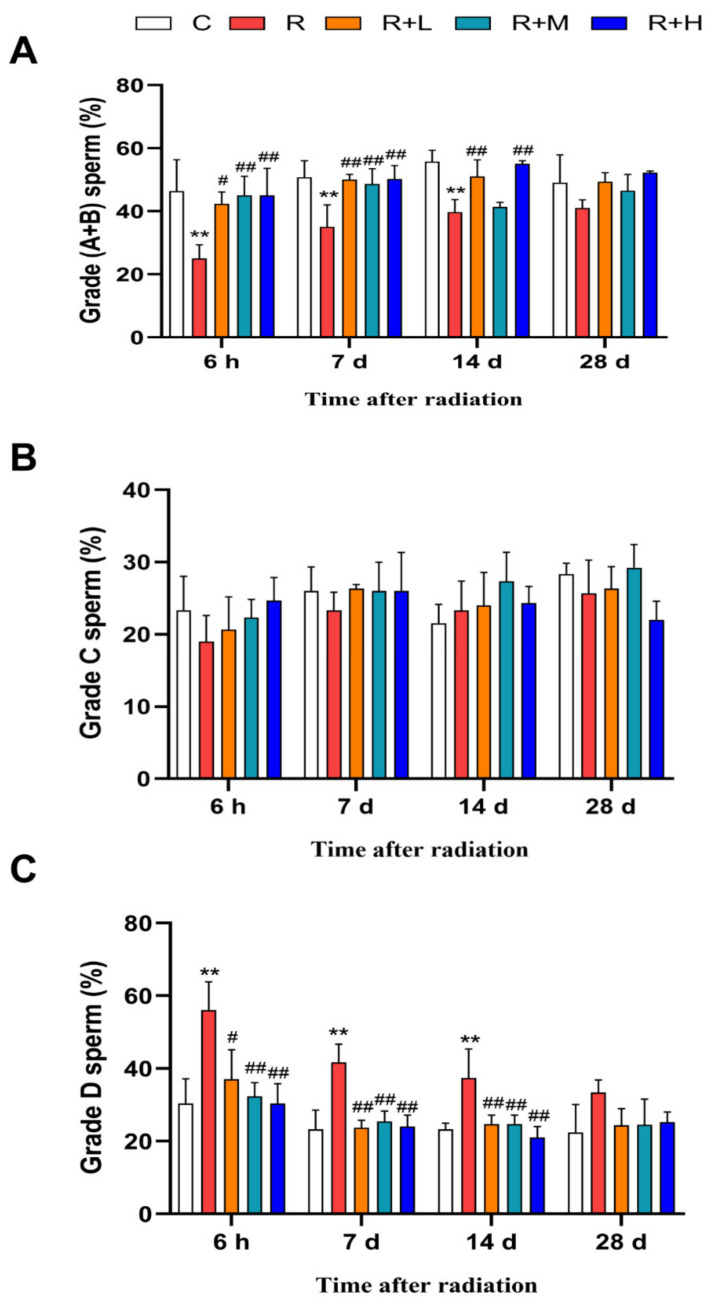
Metformin ameliorated the reduction in energy metabolism enzymes induced by microwave radiation (n = 5 for each group). (**A**) Changes in LDH content. (**B**) Changes in SDH content. (**C**) Changes in α-GLU content. * shows *p* < 0.05 for the R group compared to the C group, and ** shows *p* < 0.01; # shows *p* < 0.05 for the R + L, R + M and R + H groups compared to the R group, and ## shows *p* < 0.01; ▲ shows *p* < 0.05 for the R + M and R + H groups compared with the R + L group, ▲▲ shows *p* < 0.01; ◆ shows *p* < 0.05 for the R+ H group compared to the R + M group.

**Figure 6 ijms-24-12250-f006:**
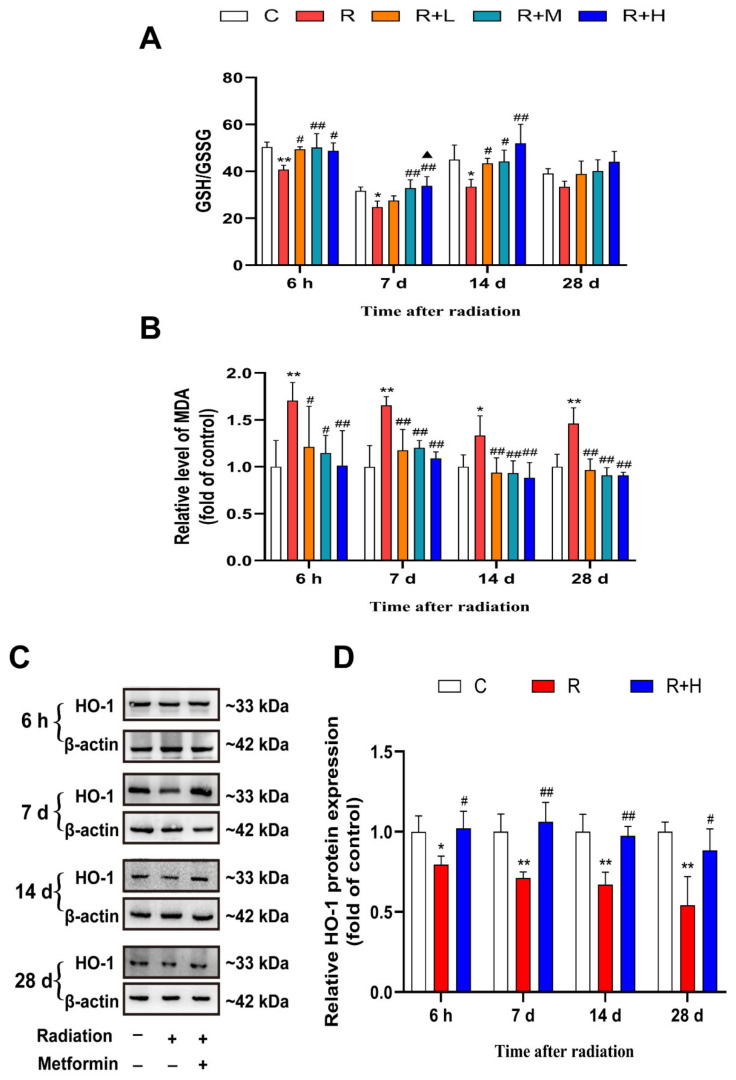
Metformin alleviated oxidative stress damage induced by microwave radiation (n = 5 for each group). (**A**) Changes in the GSH/GSSG ratio. (**B**) Changes in MDA content. (**C**) Changes in HO-1 expression. (**D**) Analysis of HO-1 expression. * shows *p* < 0.05 for the R group compared to the C group, and ** shows *p* < 0.01; # shows *p* < 0.05 for the R + L, R + M and R + H groups compared to the R group, and ## shows *p* < 0.01; ▲ shows *p* < 0.05 for the R + M and R + H groups compared with the R + L group.

**Figure 7 ijms-24-12250-f007:**
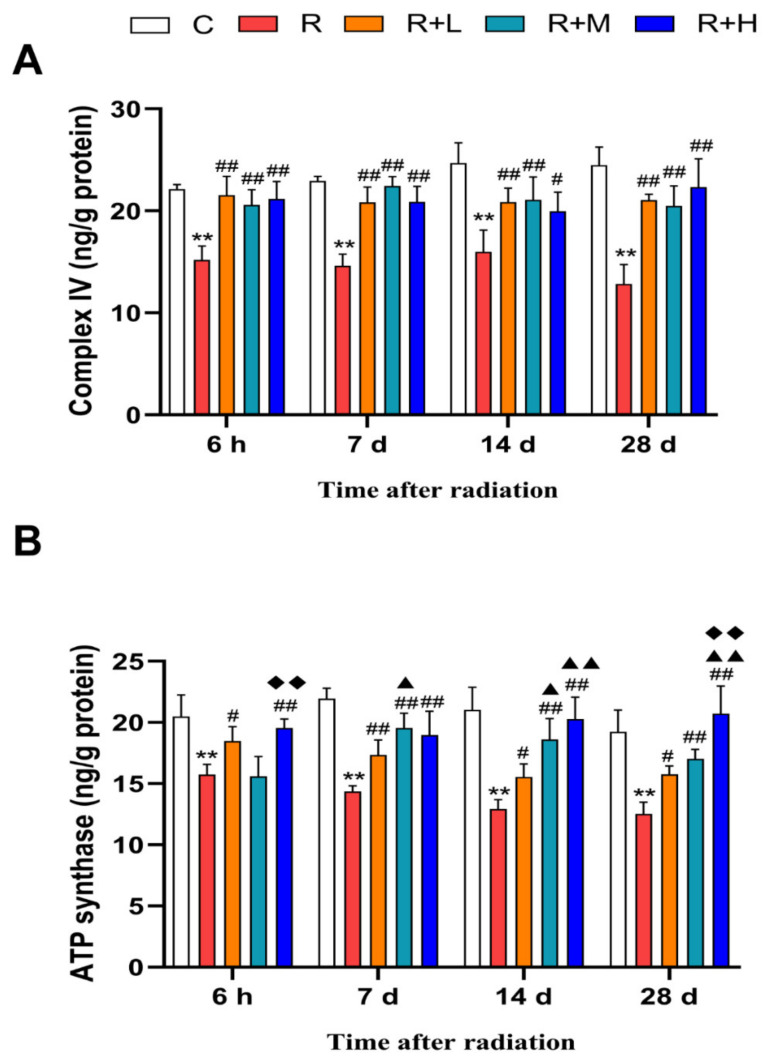
Metformin ameliorated microwave radiation-induced mitochondrial impairment (n = 5 for each group). (**A**) Changes in mitochondrial complex IV content. (**B**) Changes in ATP synthase content. ** shows *p* < 0.01 for the R group compared to the C group; # shows *p* < 0.05 for the R + L, R + M, and R + H groups compared to the R group, and ## shows *p* < 0.01; ▲ shows *p* < 0.05 for the R + M and R + H groups compared with the R + L group, and ▲▲ shows *p* < 0.01; ◆◆ shows *p* < 0.01 for the R + H group compared to the R + M group.

**Figure 8 ijms-24-12250-f008:**
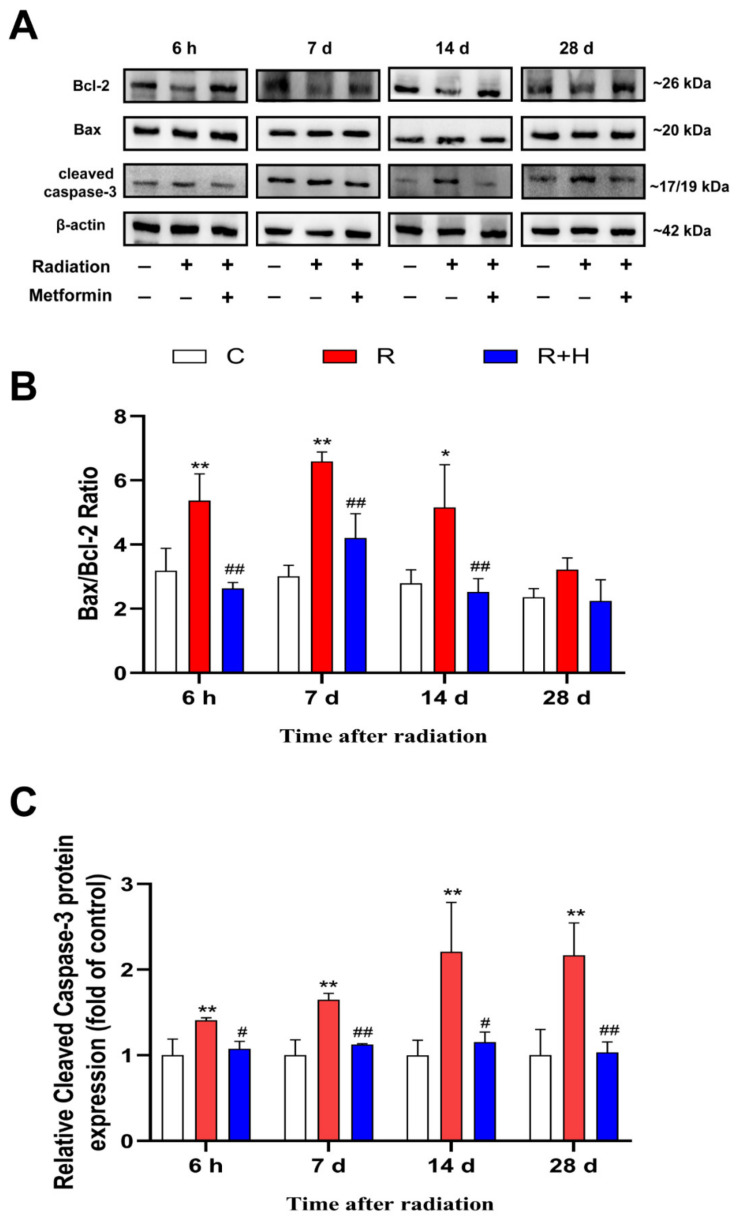
Metformin mitigated microwave radiation-induced apoptosis (n = 5 for each group). (**A**) Changes in the expression of Bcl-2, Bax, and cleaved caspase-3. (**B**) Analysis of Bax/Bcl-2, cleaved caspase-3 expression. (**C**) TUNEL staining. * shows *p* < 0.05 for the R group compared to the C group, and ** shows *p* < 0.01; # shows *p* < 0.05 for the R + H group compared to the R group, and ## shows *p* < 0.01.

**Figure 9 ijms-24-12250-f009:**
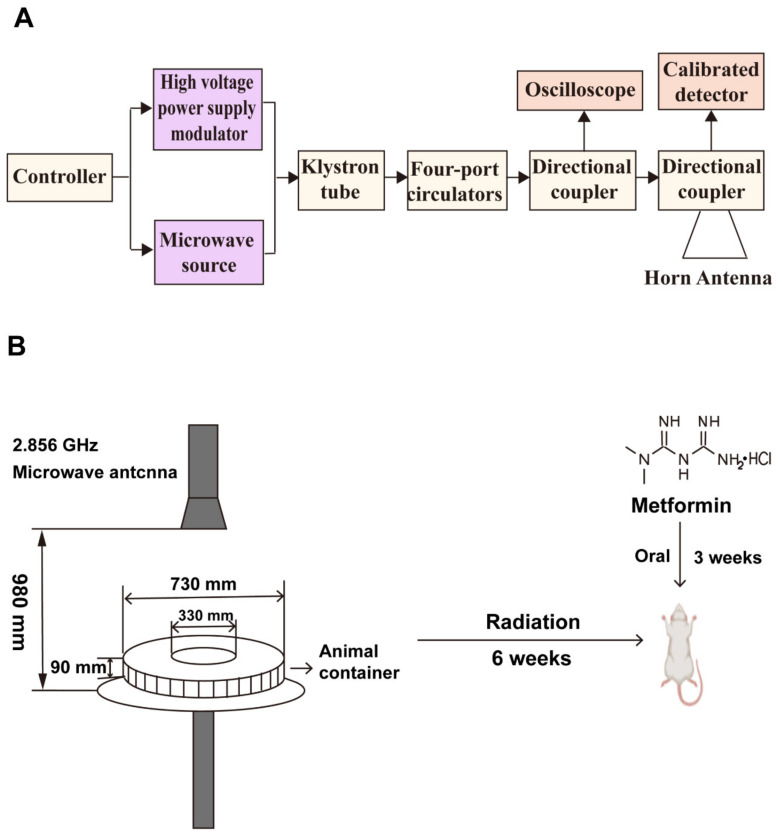
Schematic diagram of the experimental procedure and radiation apparatus. (**A**) Schematic diagram of the microwave radiation source structure. (**B**) Microwave radiation device and experimental design.

## Data Availability

The authors would like to share the detailed/raw data privately with interested researchers.

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
