# Peer review of "Metformin Ameliorates 2.856 GHz Microwave- Radiation-Induced Reproductive Impairments in Male Rats via Inhibition of Oxidative Stress and Apoptosis"

_ijms, 2023, doi:10.3390/ijms241512250_

Round 1
Reviewer 1 Report
This study demonstrates the microwave radiation-induced impairments in the male reproductive system and the mitigation of these effects by metformin, mainly by inhibition of oxidative stress and apoptosis. It is a good contribution since the authors make use of different methods to achieve the results.
In general, at the time point 28 days there were no significant results, the authors can give an explanation for that? Did the authors consider a longer period, like 10 weeks?
Moreover, the reviewer has some specific comments to improve the manuscript.
Specific comments:
1. In the Materials and Methods section:
The authors said 6- to 8- weeks old male Wistar rats, please clarify the exact age of the animals.
Ad libitum should be in italics.
More details should be given concerning Animal care and guidelines and committees.
Line 111, please verify Figures 1B-D.
In Figure 1.B Metformin was not given 3 weeks (before the last radiation), instead 14 days before and 7 after. The total is 3 weeks.
Please write by extensive ICNIRP, SAR
In line 133 the reviewer believes that is 3 µm instead of 3 mm.
In 2.9 please replace “second” by “secondary” antibody
2. In the results section:
Figure 5, for testosterone, the reviewer has some doubts regarding the statistical significance difference between the radiation group and the radiation mote metformin groups. Moreover, the error bars are bigger.
In this section in the graphs appears R, R+L, R+M, R+H, however, during the manuscript radiation is represented as MR. This should be uniformized.
The blots for HO-1 at 28 days leave some doubts.
3. The article “The protective effect of regucalcin against radiation-induced damage in testicular cells” Life Sciences Volume 164, 1 November 2016, Pages 31-41 would be a good reference to be added to the introduction/discussion.
Author Response
Reviewer #1
This study demonstrates the microwave radiation-induced impairments in the male reproductive system and the mitigation of these effects by metformin, mainly by inhibition of oxidative stress and apoptosis. It is a good contribution since the authors make use of different methods to achieve the results.
In general, at the time point 28 days there were no significant results, the authors can give an explanation for that? Did the authors consider a longer period, like 10 weeks?
Response:Thank you very much for your comment. We are highly encouraged by your positive evaluation of our work. Microwave radiation has been shown to cause damage to various organisms at different biological levels, but the damage could be recovered to some extent [1]. In the present study, some deleterious effects induced by microwave radiation were recovered at the time point 28 days, so that no significant results were found. Similar results were also found in previous studies where a 28-day exposure to microwave radiation was applied [2-4]. We do think it is interesting to perform further investigations with longer period in the future.
Moreover, the reviewer has some specific comments to improve the manuscript.
Specific comments:
- In the Materials and Methods section:
The authors said 6- to 8- weeks old male Wistar rats, please clarify the exact age of the animals.
Ad libitum should be in italics.
More details should be given concerning Animal care and guidelines and committees.
Line 111, please verify Figures 1B-D.
In Figure 1.B Metformin was not given 3 weeks (before the last radiation), instead 14 days before and 7 after. The total is 3 weeks.
Please write by extensive ICNIRP, SAR
In line 133 the reviewer believes that is 3 µm instead of 3 mm.
In 2.9 please replace “second” by “secondary” antibody
Response:Thank you for your suggestions. We have revised the manuscript according to your valuable comments as detailed below.
1) The age of the animals has been clarified to 8 weeks in the revised manuscript (Page 4, section 2.1, line 102).
2) “Ad libitum” has been corrected to the “Ad libitum” (Page 5, section 2.1, line 106).
3) The laboratory animals were handled according to the US National Institutes of Health Guide for the Care and Use of Laboratory Animals. (Page 5, section 2.1, line 107-108).
4) We have revised the description in the Figure1. Figure 1A shows the principle of microwave radiation and Figure 1B shows the radiation apparatus and experimental design (Page 5-6, section 2.2, line 134-136).
5) We have removed 'before the last radiation' from Figure 1B and revised Figure 1B accordingly (Page 6).
6) We calculated a whole-body average SAR value for rats of 10.17 W/kg. (Page 7, section 2.2, line 146-147).
7) The 3 mm in section 2.4 represents the thickness of testicular tissue dipped in wax in the embedding cassette.
8) ‘Second’ has been corrected to the ‘secondary’ (Page 8, section 2.9, line 215).
- In the results section:
Figure 5, for testosterone, the reviewer has some doubts regarding the statistical significance difference between the radiation group and the radiation mote metformin groups. Moreover, the error bars are bigger.
In this section in the graphs appears R, R+L, R+M, R+H, however, during the manuscript radiation is represented as MR. This should be uniformized.
The blots for HO-1 at 28 days leave some doubts.
Response:Thank you very much for your suggestions.
1) As described in the Material and Methods, one-way ANOVA and LSD post hoc tests were used for data statistical analysis using SPSS software. All quantitative data were expressed as mean ± standard deviation. The data of testosterone levels shown in figure 5 include a total of 25 rats with 5 rats for each group. Though the data error bars look bigger in the radiation group, the mean of the radiation group was significantly higher than the mean of the radiation + metformin group.
2) We have changed all MR in the manuscript into microwave radiation.
3) We have replaced with another representative western blots of HO-1(Page 20, Figure 7C).
- The article “The protective effect of regucalcin against radiation-induced damage in testicular cells” Life Sciences Volume 164, 1 November 2016, Pages 31-41 would be a good reference to be added to the introduction/discussion.
Response: Thanks a lot for your suggestions. We agreed this article is important. This study demonstrated the importance of the testis for male reproduction and the fact that ionising radiation could cause apoptosis in testicular tissue, and that regucalcin protected against apoptosis induced by ionising radiation. We have cited this article as references in the discussion in the revised manuscript (Page 25, line 417).
References:
- Yao, C.; Wang, H.; Sun, L.; Ren, K.; Dong, J.; Wang, H.; Zhang, J.; Xu, X.; Yao, B.; Zhou, H.; Zhao, L.; Peng, R., The Biological Effects of Compound Microwave Exposure with 2.8 GHz and 9.3 GHz on Immune System: Transcriptomic and Proteomic Analysis. Cells 2022, 11, (23).
- Li, D.; Xu, X.; Gao, Y.; Wang, J.; Yin, Y.; Yao, B.; Zhao, L.; Wang, H.; Wang, H.; Dong, J.; Zhang, J.; Peng, R., Hsp72-Based Effect and Mechanism of Microwave Radiation-Induced Cardiac Injury in Rats. Oxid Med Cell Longev 2022, 2022, 7145415.
- Wang, H.; Liu, Y.; Sun, Y.; Zhao, L.; Dong, J.; Xu, X.; Wang, H.; Zhang, J.; Yao, B.; Zhao, X.; Liu, S.; Zhang, K.; Peng, R., Changes in rat spatial learning and memory as well as serum exosome proteins after simultaneous exposure to 1.5 GHz and 4.3 GHz microwaves. Ecotoxicol Environ Saf 2022, 243, 113983.
- Zhao, L.; Yao, C.; Wang, H.; Dong, J.; Zhang, J.; Xu, X.; Wang, H.; Yao, B.; Ren, K.; Sun, L.; Peng, R., Immune Responses to Multi-Frequencies of 1.5 GHz and 4.3 GHz Microwave Exposure in Rats: Transcriptomic and Proteomic Analysis. Int J Mol Sci 2022, 23, (13).
Reviewer 2 Report
In this manuscript the authors explore the role of metformin, as antioxidant and anti-apoptotic, possible positive effects against microwaves radiation.
In my opinion this topic is relevant for the field, once human exposure to this kind of radiation and others should be a matter of concern. The addressed question is very interesting once regarding the negative and positive effects of microwaves and metformin (respectively) on male fertility.
To the best of my knowledge, I did not find a similar study. Regarding the negative effects of microwaves on male fertility and the beneficial effects of metformin in the same process can be an interesting binominal to explore.
The methodology and the results are well presented for me, maybe just two observations: - The number of animals used should be indicated in the "Microwave radiation and animal treatments" section. - Does the weight of the rats equal among all experimental groups, at the end of the experiment? Regarding the figures, in figure 3, the scale bar definition does not appear in all the panels, although it is indicated in the first and in the legend, the same for figure 4. The discussion is appropriate for the results and the main question is addressed. Also, the references are appropriate to support the manuscript.
Author Response
Reviewer #2
In this manuscript the authors explore the role of metformin, as antioxidant and anti-apoptotic, possible positive effects against microwaves radiation.
In my opinion this topic is relevant for the field, once human exposure to this kind of radiation and others should be a matter of concern. The addressed question is very interesting once regarding the negative and positive effects of microwaves and metformin (respectively) on male fertility.
To the best of my knowledge, I did not find a similar study. Regarding the negative effects of microwaves on male fertility and the beneficial effects of metformin in the same process can be an interesting binominal to explore.
The methodology and the results are well presented for me, maybe just two observations: - The number of animals used should be indicated in the "Microwave radiation and animal treatments" section.
- Does the weight of the rats equal among all experimental groups, at the end of the experiment?
Regarding the figures, in figure 3, the scale bar definition does not appear in all the panels, although it is indicated in the first and in the legend, the same for figure 4.
The discussion is appropriate for the results and the main question is addressed. Also, the references are appropriate to support the manuscript.
Response: Thank you so much for your appreciation and encouragement to us. We have revised the manuscript according to your valuable comments as detailed below.
1) The total number of animals was indicated in the section 2.1 "Animals", and the number of animals in each group has been indicated in section 2.2"Microwave radiation and animal treatments"(Page 5, line 113).
2) All animals were subjected to weighing throughout the experiments. The animals in the radiation group were found to exhibit reduced weight compared to the control group at the end of the radiation. Nonetheless, the administration of metformin efficiently mitigated these changes.
3) We have added all scale bar definitions in figure 3 and figure 4.